# Edge–Cloud Collaborative Optimization Scheduling of an Industrial Park Integrated Energy System

Gengshun Liu [1], Xinfu Song [2], Chaoshan Xin [2], Tianbao Liang [3,*], Yang Li [3] and Kun Liu [3]

1   Guoneng Xinjiang Ganquanbao Comprehensive Energy Co., Ltd., Urumqi 830019, China; 12061784@ceic.com
2   Economic and Technological Research Institute, State Grid Xinjiang Electric Power Co., Ltd., Urumqi 830002, China; yang_li06@outlook.com (X.S.); 13999864263@126.com (C.X.)
3   School of Automation Science and Engineering, Xi'an Jiaotong University, Xi'an 710049, China; yanglipiaopiao@stu.xjtu.edu.cn (Y.L.); kliu@sei.xjtu.edu.cn (K.L.)
*   Correspondence: liangtb@stu.xjtu.edu.cn

**Abstract:** Due to the large proportion of China's energy consumption used by industry, in response to the national strategic goal of "carbon peak and carbon neutrality" put forward by the Chinese government, it is urgent to improve energy efficiency in the industrial field. This paper focuses on the optimization of an integrated energy system with supply–demand coordination in an industrial park. This optimization is formulated as a "node-flow" model. Within the model, each node is designed according to the objective function of its own operation and the energy coupling relationship. The flow model is designed based on the energy flow interaction relationship between each node. Based on the "node-flow" model, an edge–cloud information interaction mechanism based on energy transfer balance between nodes is proposed to describe the way the system interacts with information, and a distributed iterative optimization algorithm based on edge–cloud collaboration is designed to realize the optimization decision of each node. The performance of the method proposed in this paper is demonstrated using a practical case study of an industrial park integrated energy system in Xinjiang. The results show that the proposed model can effectively improve the utilization efficiency of multi-energy synergy and complementation in the industrial park, and the proposed algorithm can shorten the solution time by more than 50% without significantly affecting the accuracy of the solution.

**Keywords:** industrial park; integrated energy system; distributed iterative algorithm; optimal operation



## 1. Introduction

### 1.1. Motivation

Recently, China's industrial energy consumption has accounted for about 65% of the total energy consumption by the whole of society [1]. In this context, carbon emissions from industrial parks can reach 31% of the country's total emissions [2]. In response to the national strategic goal of "carbon peak and carbon neutral" put forward by the Chinese government, it is important to improve energy efficiency in the industrial sector to realize this strategic goal [3]. However, in industrial energy systems, different items of energy supply equipment are relatively independent, and the overall energy efficiency is not high [4]. Therefore, research on multi-energy synergy and energy efficiency improvement of integrated energy systems in industrial parks is of great significance.

### 1.2. Literature Review

Reducing the negative environmental impacts of industries is a major challenge, both in advanced and emerging economies [5]. Direct emissions mainly originate from fuel combustion and industrial processes, while indirect emissions are primarily from energy production (electricity and heat) [6]. Improvements in energy efficiency and a

greater deployment of renewable energy are considered as essential for a low-carbon transition [7]. Industrial parks, as economic engines for many regions [8], have high energy consumption and play an important role in the local target of carbon reduction and energy conservation [9–12].

Under the gigantic topic of "carbon peak and carbon neutral", smart solutions for sustainable and low-carbon transition are needed and have attracted increasing attention. Part of the research has focused on the modeling of industrial parks. Wu et al. studied an individualized electricity and thermal coupled pricing strategy for consumers with different demand profiles in an industrial park and presents a dynamic pricing mechanism for the industrial park with demand response programs [13,14]. Cao et al. proposed a reputation factor pricing strategy for a shared energy storage station (SESS) for industrial parks that enables the SESS to allocate energy fairly and efficiently under limited power constraints [15]. Yang et al. constructed an industrial park microgrid integrated energy system model to improve the energy efficiency of an industrial park [16]. Hu et al. proposed a structure for an integrated energy system for a coal mine, considering the economic cost, carbon transaction cost for environment protection, and degree of customer dissatisfaction with reducible and translational load [17]. Wang et al. proposed an operation scheduling method for a park-level integrated electric and heating system considering carbon trading for reducing carbon emissions and improving energy efficiency [18]. Zhu et al. proposed a regional integrated energy systems energy management strategy based on stepped utilization of energy to further minimize the daily cost of the industrial park and make full use of the energy [19]. Choobineh et al. proposed an alternative approach based on multi-objective optimization to maximize the collective benefits of a group of industrial enterprises [20]. Yang et al. used a pinch point algorithm to analyze various loads in energy storage systems found in autonomous microgrids [21]. Wang et al. introduces an optimization strategy tailored to clustered factories, considering the incorporation of carbon trading and supply chain integration throughout the entire production process of each factory [22]. Xu et al. proposed a bi-level multi-objective model for industrial park distributed energy configuration optimization to deal with extreme events [23]. Ning et al. proposed a differential pricing strategy for the potential game of IPEO to achieve clean and efficient operation of industrial electric-heat energy systems [24].

Another part of the research has focused on model solving. Guo et al. constructed a regional integrated energy system model considering demand response to solve the problem that the existing evaluation system for the energy system was not comprehensive [25]. Gu et al. proposed a bi-level low-carbon economic dispatch model for the industrial park and solved it using a nonlinear primal–dual path-following interior-point method [26]. Xing et al. presented an augmented $\varepsilon$-constraint method to solve the multi-objective optimization model for distributed energy systems in an industrial park [27]. Oskouei et al. established a hierarchical optimization structure for solving the demand response aggregator self-scheduling problem by identifying the behavior of industrial consumers, which corresponds with the demand response aggregator, by means of the load disaggregation approach [28]. Ge et al. proposed a dual-level scheduling model of the microgrid system including day-ahead and real-time scheduling and solved it using an improved particle swarm optimization algorithm [29]. Liberona et al. proposed a methodology to design efficient eco-industrial parks that are also robust to daily uncertain variations of the nominal operation of the enterprises [30]. Yan et al. presented a dynamic recognition technology to recognize the cluster to which the intra-day load curve belongs and provides a cost-effective solution for energy storage system operation in an industrial park [31].

*1.3. Contributions*

As indicated by the literature review, optimal dispatching of energy systems in industrial parks can effectively improve energy efficiency, but there are still many challenges in achieving coordinated optimization of supply and demand in energy systems in reality. First, there are industrial enterprises with diverse energy demands in industrial parks, as

well as energy supply equipment with diverse energy supply forms and geographically dispersed distribution. It is difficult to obtain the status of various equipment in industrial parks accurately and quickly. Second, various energy conversion and storage devices in industrial parks cause spatio-temporal multi-scale coupling of electricity, heat, gas, and other energy sources in the system. It is particularly important to establish a refined multi-energy coupling model of system supply and demand. Third, due to the above two challenges in large-scale industrial parks, it is difficult to obtain the system operation strategy effectively and quickly through centralized solving algorithms. It is of great significance to analyze the correlation between the information network and the energy supply and demand system between system nodes, establish an information interaction mechanism, and design an efficient distributed collaborative optimization algorithm.

In order to solve the above challenges, this paper focuses on the sustainable development of industrial parks under the background of "carbon peak and carbon neutrality". Specifically, the optimization of an integrated energy system with supply-demand coordination in an industrial park is studied. This paper focuses on improving the efficiency of the cooperative operation of energy supply and demand equipment in industrial parks. The main contributions are as follows:

(1) The "node-flow" model of the industrial park is established. Therein, various energy equipment and industrial enterprises in the system are defined as edge nodes, and each node model is designed according to the objective function of its own operation and the energy coupling relationship. The flow model is designed based on the energy flow interaction relationship between each node model. Combining each node model and the flow model, the complex coupling relationship between supply and demand of the integrated energy system in the industrial park and the energy interaction relationship between each node can be accurately described.

(2) Based on the "node-flow" model, an edge–cloud information interaction mechanism is proposed to ensure the energy interaction balance between nodes. Based on this mechanism, a distributed iterative optimization algorithm was designed to optimize the operation strategy of each node, achieving efficient and optimized operation of the integrated energy system in the industrial park.

(3) This paper verifies the performance of the proposed method based on actual industrial park integrated energy system operating data.

The rest of this paper is organized as follows: the "node-flow" model of the collaborative supply and demand optimization problem of the integrated energy system in the industrial park is described in Section 2. The edge–cloud information interaction mechanism and the distributed iterative optimization algorithm based on edge–cloud collaboration are introduced in Section 3. The performance of the proposed method is demonstrated using numerical case tests in Section 4. Finally, conclusions are given in Section 5.

## 2. Problem Description and Model Formulation

This paper studies the operation optimization problem in the integrated energy system of an industrial park. The industrial park system under study is shown in Figure 1, and the correlation of multiple energy and information flows within the system is considered. With the goal of minimizing the operating cost of the industrial park, the various links of supply, storage, and demand within the system are coordinated to satisfy the demand of industrial enterprises for multiple energy sources and to achieve the optimal operational scheduling of the system. In this paper, various types of energy supply and conversion equipment as well as industrial enterprises in the industrial park are defined as nodes, the energy flow and information flow interacting among nodes are defined as edges, and the graph network model $G = \langle V, E \rangle$ of the system is obtained, where $V$ denotes the set of all nodes and $E$ is the set of directed edges connecting the nodes $E \subseteq V \times V$.

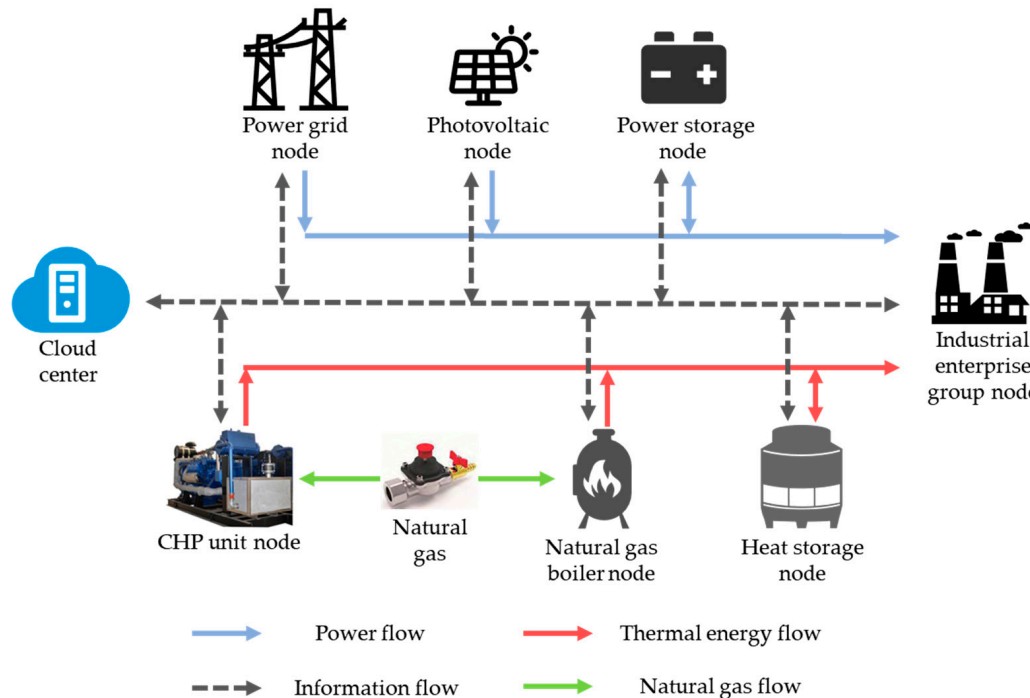

**Figure 1.** Framework of multi-energy flow information flow in industrial parks.

Based on the energy type, the system is split into a three-layer structure of electric energy flow network, heat energy flow network, and information flow network, as in Figure 1. Seven types of nodes are defined, which are power grid node, photovoltaic node, natural-gas-driven combined heat and power (CHP) generation unit node, power storage node constituted by storage batteries, heat storage node, natural gas boiler node, and industrial enterprise group node.

In order to solve the problem of solution complexity caused by the large number of nodes in large-scale industrial parks, a node-flow model is established and the centralized optimization problem is divided into several sub-optimization problems. The optimization goal of the centralized problem is to minimize the total cost of electricity and natural gas, while the optimization goal of each node problem is to add the energy interaction cost minimization on this basis. Thus, the optimization objectives of each subproblem are independent. In addition, the node model belonging to each "edge node" describes the dynamic transformation relationship between various energies at each node, and the flow model describes the interaction process of electric energy, heat energy, and information between nodes, which ensures the energy balance between nodes and the safe exchange of information. The above two types of energy balance ensure the overall supply and demand balance of the system. The specific model is introduced in detail below and the nomenclature is shown in Table 1.

**Table 1.** The nomenclature used in this paper.

| Nomenclature | | | |
|---|---|---|---|
| Abbreviation | | | |
| *pg* | Power grid node | *pv* | Photovoltaic node |
| *CHP* | Combined heat and power/combined heat and power unit node | *se* | Electrical power storage node |
| *R* | Industrial enterprise user group node | *gb* | Natural gas boiler node |
| *sh* | Heat storage node | SOC | State of charge of the battery |

**Table 1.** *Cont.*

| **Nomenclature** | | | |
|---|---|---|---|
| Parameters | | | |
| $K$ | The total length of the scheduling cycle | $c_k^b / c_k^s$ | Buying/selling electricity prices of the power grid at time $k$ |
| $w_k^{m \to n}$ | Electrical line loss from node $m$ to node $n$ in the system at time $k$ | $LC$ | Line capacity limit |
| $E^b / E^s$ | The upper limit of the system's buying/selling power | $\tau$ | The time stage length of the optimization model |
| $P_k^{pv}$ | Power generation power of the photovoltaic node at time $k$ | $C_k^{ng}$ | Unit price of natural gas at time $k$ |
| $\overline{x^c}$ | The upper limit of the load factor of CHP unit | $\Delta x_{max}^c$ | The upper limit of the load factor's change of CHP unit |
| $P^c$ | The rated load of CHP unit | $a, b, c, d$ | CHP unit parameters related to rated power |
| $f^{m \to n}$ | Heat line loss from node $m$ to node $n$ in the system | $E$ | The maximum storage capacity of the power storage node |
| $\mu^b$ | The power attenuation coefficient | $\alpha^{bc} / \beta^{bd}$ | The charge/discharge coefficients of the power storage node |
| $\underline{V^b} / \overline{V^b}$ | The upper/lower limits of the battery state of charge of the power storage node | $V_0^b$ | The initial state of battery charge |
| $\underline{p^{bc}} / \overline{p^{bc}}$ | The upper and lower limits of the charging power of the power storage node | $\underline{p^{bd}} / \overline{p^{bd}}$ | The upper and lower limits of the discharging power of the power storage node |
| $\overline{V^h}$ | The upper limit of the heat storage capacity of the heat storage node | $\mu^h$ | The heat attenuation coefficient of the heat storage tank |
| $V_0^h$ | The initial state of the heat storage tank | $\overline{q^{in}} / \overline{q^{out}}$ | The upper limit of the heat storage node's charge/discharge heat per unit time |
| $z_k^{in} / z_k^{out}$ | Binary decision variable for the charging/discharging of the heat storage node at time $k$. | $H^{ng}$ | The calorific value of natural gas |
| $\eta^{gb}$ | The thermal energy conversion coefficient of the natural gas boiler | $Q_{max}^{gb}$ | The upper limit of heat production of natural gas boilers |
| Variables | | | |
| $e_k^b / e_k^s$ | Buying/selling power of the system at time $k$ | $\lambda_k^{m \to n}$ | Electrical line interaction cost from node $m$ to node $n$ in the system at time $k$ |
| $e_k^{m \to n}$ | Power delivered by node $m$ to node $n$ in the system at time $k$ | $e_k^{n \leftarrow m}$ | Power received by node $n$ from node $m$ in the system at time $k$ |
| $z_k^{pg \to se} / z_k^{pg \leftarrow se}$ | Binary decision variable of the interaction between the power grid node and the power storage node at time $k$ | $z_k^b / z_k^s$ | The binary decision variable of the buying/selling power of the system at time $k$ |
| $V_k^{ng\_CHP}$ | The natural gas consumption of the CHP unit at time $k$ | $P_k^{CHP}$ | The electric output power of the CHP unit at time $k$ |
| $z_k^c$ | The decision variable for turning on and off CHP unit at time $k$ | $x_k^c$ | The load factor of CHP unit at time $k$ |
| $Q_k^{CHP}$ | The thermal energy output of CHP unit at time $k$ | $q_k^{m \to n}$ | The thermal energy transmitted from node $m$ to node $n$ in the system at time $k$. |
| $q_k^{n \leftarrow m}$ | The thermal energy received from node $m$ by node $n$ in the system at time $k$ | $\mu^{m \to n}$ | Heat line interaction cost from node $m$ to node $n$ in the system |
| $s_k^b$ | The battery state of charge of the power storage node at time $k$ | $z_k^{bc} / z_k^{bd}$ | The binary decision variable for charging/discharging of the power storage node at time $k$ |

**Table 1.** *Cont.*

| Nomenclature | | | |
|---|---|---|---|
| $p_k^{bc}, p_k^{bd}$ | The charging/discharging power of the power storage node at time $k$ | $q_k^h$ | The heat storage state of the heat storage node at time $k$ |
| $V_k^{ng\_gb}$ | The natural gas consumption of the natural gas boiler node at time $k$ | | |

### 2.1. Power Grid Node Model

The optimization goal of the power grid node is to minimize the power interaction cost of the power grid during the dispatch period and the energy interaction cost between its connected nodes, while satisfying the energy conservation and node operation constraints. The objective function of the model is as follows:

$$\min \sum_{k=1}^{K} (c_k^b e_k^b - c_k^s e_k^s + \lambda_k^{pv \to pg} e_k^{pg \leftarrow pv} + \lambda_k^{CHP \to pg} e_k^{pg \leftarrow CHP} + \lambda_k^{se \to pg} e_k^{pg \leftarrow se}$$
$$- \lambda_k^{pg \to se} w^{pg \to se} e_k^{pg \to se} - \lambda_k^{pg \to R} w^{pg \to R} e_k^{pg \to R}) \tag{1}$$

where $K$ is the total length of the scheduling cycle. $c_k^b$ and $c_k^s$, respectively, are the buying and selling electricity prices of the power grid at time $k$. $e_k^b$ and $e_k^s$ are, respectively, the buying power and selling power of the system at time $k$. $\lambda_k^{m \to n}$ and $w_k^{m \to n}$, respectively, represent the line interaction cost and line loss from node $m$ to node $n$ in the system at time $k$. $e_k^{m \to n}$ and $e_k^{n \leftarrow m}$, respectively, represent the power delivered by node $m$ to node $n$ and the power received by node $n$ from node $m$ in the system at time $c_w$. The superscript $pg$, $pv$, $CHP$, $se$, $R$ indicate power grid node, photovoltaic node, CHP unit node, power storage node, and industrial enterprise user group node, respectively.

The objective function consists of two parts. One part is the power interaction cost of the power grid, consisting of the first two items of Formula (1), and the other part is the cost of energy interaction with other nodes in the system, consisting of the last five items of Formula (1).

The power grid node model includes the operational constraints of the node. Specifically, there are energy conservation constraints and electricity trading constraints with the power grid. The details are as follows:

$$e_k^b + e_k^{pg \leftarrow pv} + e_k^{pg \leftarrow CHP} + e_k^{pg \leftarrow se} = e_k^s + e_k^{pg \to se} + e_k^{pg \to R} \tag{2}$$

$$0 \le e_k^{pg \leftarrow se} \le z_k^{pg \leftarrow se} \cdot LC \tag{3}$$

$$0 \le e_k^{pg \to se} \le z_k^{pg \to se} \cdot LC \tag{4}$$

$$z_k^{pg \to se} + z_k^{pg \leftarrow se} \le 1 \tag{5}$$

$$0 \le e_k^b \le z_k^b \cdot E^b \tag{6}$$

$$0 \le e_k^s \le z_k^s \cdot E^s \tag{7}$$

$$z_k^b + z_k^s \le 1 \tag{8}$$

where $LC$ is the line capacity limit. $E^b$ and $E^s$ are, respectively, the upper limit of the system's buying and selling power. $z_k^{pg \to se}$ and $z_k^{pg \leftarrow se}$ are the binary decision variable of the interaction between the power grid node and the energy storage node at time $k$. $z_k^b$ and $z_k^s$ is the binary decision variable of the buying and selling power of the system at time $k$.

Formula (2) is the node energy conservation constraint, Formulas (3)–(5) are the line transmission constraint between the distribution network node and the storage node,

Formulas (6)–(8) are the upper and lower limit constraints of the grid interactive power and grid operating status constraints.

### 2.2. Photovoltaic Node Model

This paper takes photovoltaic nodes as an example to represent the renewable energy included in a general integrated energy system. The optimization goal is to minimize the energy interaction cost of its connected nodes during operation scheduling. The objective function of the model is as follows:

$$\min \sum_{k=1}^{K} \left( -\lambda_k^{pv \to pg} w^{pv \to pg} e_k^{pv \to pg} - \lambda_k^{pv \to se} w^{pv \to se} e_k^{pv \to se} - \lambda_k^{pv \to R} w^{pv \to R} e_k^{pv \to R} \right) \quad (9)$$

Since the photovoltaic node has no energy consumption and cost, it only transmits energy outwards. The objective function of photovoltaic node model only consists of one part, that is the energy interaction cost with other nodes in the system.

$$P_k^{pv} \cdot \tau = e_k^{pv \to pg} + e_k^{pv \to se} + e_k^{pv \to R} \quad (10)$$

where $\tau$ is the time stage length of the optimization model. $P_k^{pv}$ is the power generation power of the photovoltaic node at time $k$. Formula (10) is the energy conservation constraint of the photovoltaic node, that is, the energy generated by the photovoltaic node in the time period is equal to the energy transmitted to other nodes.

### 2.3. CHP Node Model

The CHP unit consumes natural gas and generates thermal energy and electric energy, coupling the thermal energy flow and electric energy flow in the system. Its optimization goal is to minimize the cost of natural gas and the cost of energy interaction with connected nodes during the dispatch period. The objective function of the model is as follows:

$$\min \sum_{k=1}^{K} \left( C_k^{ng} V_k^{ng\_CHP} - \lambda_k^{CHP \to pg} w^{CHP \to pg} e_k^{CHP \to pg} - \lambda_k^{CHP \to se} w^{CHP \to se} e_k^{CHP \to se} \right.$$
$$\left. -\lambda_k^{CHP \to R} w^{CHP \to R} e_k^{CHP \to R} - \mu_k^{CHP \to sh} f^{CHP \to sh} q_k^{CHP \to sh} - \mu_k^{CHP \to R} f^{CHP \to R} q_k^{CHP \to R} \right) \quad (11)$$

where $C_k^{ng}$ is the natural gas unit price at time $k$. $V_k^{ng\_CHP}$ is the natural gas consumption of the CHP unit at time $k$.

The objective function consists of two parts. One part is natural gas purchase cost, consisting of the first items of Formula (11), and the other part is the cost of energy interaction with other nodes in the system, consisting of the last five items of Formula (11).

The CHP unit produces heat and electricity by consuming natural gas. Its natural gas consumption, electricity, and heat energy are related as follows [32]:

$$P_k^{CHP} = P^c \cdot x_k^c \quad (12)$$

$$Q_k^{CHP} = (a x_k^c + b z_k^c) \tau \quad (13)$$

$$V_k^{ng\_CHP} = (c x_k^c + d z_k^c) \tau \quad (14)$$

where $z_k^c$ is the decision variable for turning on and off the CHP unit at time $k$. $x_k^c$ is the load factor of CHP unit at time $k$. $Q_k^{CHP}$ is the thermal energy output of the CHP unit at time $k$. $P^c$ is the rated load of the CHP unit. $a, b, c, d$ are unit parameters related to rated power, and can be obtained through linear fitting of actual operating data [32]. In addition, during operation, CHP units are also constrained by energy conservation constraints and operating output constraints. The details are as follows:

$$x_k^c \le z_k^c \cdot \overline{x^c} \quad (15)$$

$$\left|x_{k-1}^c - x_k^c\right| \le z_k^c \cdot \Delta x_{\max}^c \tag{16}$$

$$P_k^{CHP} \cdot \tau = e_k^{CHP \to pg} + e_k^{CHP \to se} + e_k^{CHP \to R} \tag{17}$$

$$Q_k^{CHP} = q_k^{CHP \to sh} + q_k^{CHP \to R} \tag{18}$$

where $\overline{x^c}$ is the upper limit of the load factor of CHP unit. $\Delta x_{\max}^c$ is the upper limit of the load factor change of the CHP unit. $q_k^{m \to n}$ and $q_k^{n \gets m}$ respectively represent the thermal energy transmitted from node $m$ to node $n$ and the thermal energy received from node $m$ by node $n$ in the system at time $k$. $\mu^{m \to n}$ and $f^{m \to n}$, respectively, represent the line interaction loss and line loss from node $m$ to node $n$ in the system at time $k$. The superscript *sh* indicates the heat storage node. Formula (15) gives the load rate and start-up constraints of the cogeneration unit. Formula (16) is the climbing constraint of the cogeneration unit. Formulas (17) and (18) are the energy conservation constraints of the CHP unit nodes.

### 2.4. Power Storage Node Model

The power storage node can store and supply power to the system. Its optimization goal is to minimize the cost of energy interaction with connected nodes during operation scheduling. The objective function of the model is as follows:

$$\min \sum_{k=1}^{K} (\lambda_k^{pg \to se} e_k^{se \gets pg} + \lambda_k^{CHP \to se} e_k^{se \gets CHP} + \lambda_k^{pv \to se} e_k^{se \gets pv} - \lambda_k^{se \to pg} w^{se \to pg} e_k^{se \to pg} \\ - \lambda_k^{se \to R} w^{se \to R} e_k^{se \to R}) \tag{19}$$

The objective function of the power storage node model consists of only one part, that is, the cost of energy interaction with other nodes in the system. The power storage node must comply with the operating constraints of the energy storage system [33], so the following formulas are shown:

$$s_{k+1}^b \cdot E = s_k^b \cdot E \cdot \mu^b + [\alpha^{bc} p_k^{bc} - \frac{p_k^{bd}}{\beta^{bd}}] \cdot \tau \tag{20}$$

$$s_0^b = s_K^b = V_0^b \tag{21}$$

$$\underline{V^b} \le s_k^b \le \overline{V^b} \tag{22}$$

$$z_k^{bc} \cdot \underline{p^{bc}} \le p_k^{bc} \le z_k^{bc} \cdot \overline{p^{bc}} \tag{23}$$

$$z_k^{bd} \cdot \underline{p^{bd}} \le p_k^{bd} \le z_k^{bd} \cdot \overline{p^{bd}} \tag{24}$$

$$z_k^{bc} + z_k^{bd} \le 1 \tag{25}$$

where $s_k^b$ is the battery state of charge of the power storage node at time $k$. $E$ is the maximum storage capacity of the power storage node. $\mu^b$ is the power attenuation coefficient. $\alpha^{bc}, \beta^{bd}$ are the charge and discharge coefficients of the power storage node, respectively. $\underline{V^b}, \overline{V^b}$ are the upper and lower limits of the battery state of charge of the power storage node, respectively. $V_0^b$ is the initial state of battery charge. $\underline{p^{bc}}, \overline{p^{bc}}, \underline{p^{bd}}, \overline{p^{bd}}$ are the upper and lower limits of the charging and discharging power of the power storage node, respectively. $z_k^{bc}, z_k^{bd}$ are the binary decision variable for charging and discharging of the power storage node at time $k$. $p_k^{bc}, p_k^{bd}$ are the charging and discharging power of the power storage node at time $k$. Formula (20) represents the constraint of the state of charge (SOC) change of the battery. Formulas (21) and (22) represent the initial and final states of the battery, and the upper and lower limits of the battery. Formulas (23)–(25) indicate that the battery cannot be charged and discharged at the same time.

In addition, the operation of the node must meet the constraint of energy conservation, so the following formulas are shown:

$$p_k^{bc} \cdot \tau = e_k^{se \leftarrow pg} + e_k^{se \leftarrow CHP} + e_k^{se \leftarrow pv} \tag{26}$$

$$p_k^{bd} \cdot \tau = e_k^{se \rightarrow pg} + e_k^{se \rightarrow R} \tag{27}$$

### 2.5. Heat Storage Node Model

The heat storage node uses a hot water storage tank to store thermal energy. The operating cost of this node is the cost of energy interaction with the connected nodes. The objective function of the model is as follows:

$$\min \sum_{k=1}^{K} (\mu_k^{CHP \rightarrow sh} q_k^{sh \leftarrow CHP} + \mu_k^{gb \rightarrow sh} q_k^{sh \leftarrow gb} - \mu_k^{sh \rightarrow R} f^{sh \rightarrow R} q_k^{sh \rightarrow R}) \tag{28}$$

The objective function of the heat storage node model consists of only one part, that is, the cost of energy interaction with other nodes in the system. The heat storage node must comply with the operating constraints of the heat storage unit [34], so the following formulas are shown:

$$q_{k+1}^h = q_k^h \cdot \mu^h + q_k^{in} - q_k^{out} \tag{29}$$

$$0 \le q_k^h \le \overline{V^h} \tag{30}$$

$$0 \le q_k^{in} \le \overline{q^{in}} \cdot z_k^{in} \tag{31}$$

$$0 \le q_k^{out} \le \overline{q^{out}} \cdot z_k^{out} \tag{32}$$

$$z_k^{in} + z_k^{out} \le 1 \tag{33}$$

$$q_0^h = q_K^h = V_0^h \tag{34}$$

where $q_k^h$ is the heat storage capacity of the heat storage node at time $k$. $\overline{V^h}$ is the upper limit of the heat storage capacity of the heat storage node. $\mu^h$ is the heat attenuation coefficient of the heat storage tank. $V_0^h$ is the initial state of the heat storage tank. $\overline{q^{in}}, \overline{q^{out}}$ are the upper and lower limit of the heat storage node's charge and discharge heat per unit time. $z_k^{in}, z_k^{out}$ are the binary decision variable for the charging and discharging heat of the heat storage node at time $k$. Formula (29) represents the energy change of the heat storage unit. Formula (30) represents the upper and lower limits of the heat storage unit. Formulas (31)–(33) indicate that the heat storage unit cannot be charged and discharged at the same time. Formula (34) indicates the initial and final states of the heat storage unit.

In addition, the operation of the node must meet the constraint of energy conservation, so the following formulas are shown:

$$q_k^{in} = q_k^{sh \leftarrow CHP} + q_k^{sh \leftarrow gb} \tag{35}$$

$$q_k^{out} = q_k^{sh \rightarrow R} \tag{36}$$

where the superscript $gb$ indicates the natural gas boiler node.

### 2.6. Natural Gas Boiler Node Model

The natural gas boiler node uses natural gas to generate heat energy. The operating cost of this node is the energy interaction cost with the connected nodes. The objective function of the model is as follows:

$$\min \sum_{k=1}^{K} (C_k^{ng} V_k^{ng\_gb} - \mu_k^{gb \to sh} f^{gb \to sh} q_k^{gb \to sh} - \mu_k^{gb \to R} f^{gb \to R} q_k^{gb \to R}) \qquad (37)$$

The objective function consists of two parts. One part is natural gas purchase cost, consisting of the first items of Formula (37), and the other part is the cost of energy interaction with other nodes, consisting of the last two items of Formula (37). In addition, the natural gas boiler node produces heat by consuming natural gas. Its natural gas consumption and heat energy are related as follows:

$$Q_k^{gb} = \eta^{gb} H^{ng} V_k^{ng\_gb} \qquad (38)$$

$$0 \leq Q_k^{gb} \leq Q_{\max}^{gb} \qquad (39)$$

$$Q_k^{gb} = q_k^{gb \to sh} + q_k^{gb \to R} \qquad (40)$$

where $V_k^{ng\_gb}$ is the natural gas consumption at time $k$. $Q_k^{gb}$ is the heat production of the natural gas boiler at time $k$. $H^{ng}$ and $\eta^{gb}$ are the calorific value of natural gas and the thermal energy conversion coefficient of the natural gas boiler, respectively. $Q_{\max}^{gb}$ is the upper limit of heat production of the natural gas boiler. Formula (38) represents the relationship between natural gas consumption and the heat energy of the node. Formula (39) represents the upper and lower limits of natural gas boiler output. Formula (40) represents the energy conservation constraint of the node.

### 2.7. Industrial Enterprise Group Node Model

The industrial enterprise group node is the main energy-consuming part of the system. The operating cost of this node is the cost of energy interaction with the connected nodes. The objective function of the model is as follows:

$$\min \sum_{k=1}^{K} (\lambda_k^{pg \to R} e_k^{R \leftarrow pg} + \lambda_k^{CHP \to R} e_k^{R \leftarrow CHP} + \lambda_k^{se \to R} e_k^{R \leftarrow se} + \lambda_k^{pv \to R} e_k^{R \leftarrow pv}$$
$$+ \mu_k^{CHP \to R} q_k^{R \leftarrow CHP} + \mu_k^{sh \to R} q_k^{R \leftarrow sh} + \mu_k^{gb \to R} q_k^{R \leftarrow gb}) \qquad (41)$$

The objective function of industrial enterprise group node consists of only one part, that is, the cost of energy interaction with other nodes in the system. It needs to comply with the following energy conservation constraints:

$$e_k^{R \leftarrow pg} + e_k^{R \leftarrow CHP} + e_k^{R \leftarrow se} + e_k^{R \leftarrow pv} = \sum_{i=1}^{N} e_k^{R,i} \qquad (42)$$

$$q_k^{R \leftarrow CHP} + q_k^{R \leftarrow sh} + q_k^{R \leftarrow gb} = \sum_{i=1}^{N} q_k^{R,i} \qquad (43)$$

where the superscript $i$ represents the industrial enterprise node label. The industrial enterprise node considered in this article refers to an area composed of one or more industrial enterprises, which $N$ is the total number of industrial enterprise nodes. Formulas (42) and (43) are the energy conservation constraints of the industrial enterprise group node.

### 2.8. Flow Model

This section establishes a flow model for interaction between nodes based on the above node model. It is specifically divided into electrical and thermal energy interactions. The model is as follows:

$$w^{m \to n} e_k^{m \to n} = e_k^{n \leftarrow m} \qquad (44)$$

$$0 \leq e_k^{m \to n} \leq LC, 0 \leq e_k^{n \leftarrow m} \leq LC \tag{45}$$

$$f^{m \to n} q_k^{m \to n} = q_k^{n \leftarrow m} \tag{46}$$

$$0 \leq q_k^{m \to n} \leq LC, 0 \leq q_k^{n \leftarrow m} \leq LC \tag{47}$$

$$m, n \in M = (pg, pv, CHP, se, sh, gb, R) \tag{48}$$

Formulas (44)–(47) are the electric energy and thermal energy transmission constraints, which ensure the supply and demand balance between system nodes and the overall balance. In this paper, a simplified model of power network and heat network is considered, and the inter-node flow loss is used to describe the power and heat network between nodes.

## 3. Distributed Iterative Optimization Algorithm

The optimization problem of integrated energy systems in industrial parks contains a large number of decision variables and the decision space is huge. Moreover, it is difficult for the system to obtain all the information for each part accurately and in a timely manner. Therefore, this paper proposes an information interaction mechanism between edge nodes and the cloud based on the node model and traffic model. An information processing center is set up in the cloud to collect equipment information from each node in the industrial park and coordinate its decision making, as shown in Figure 2.

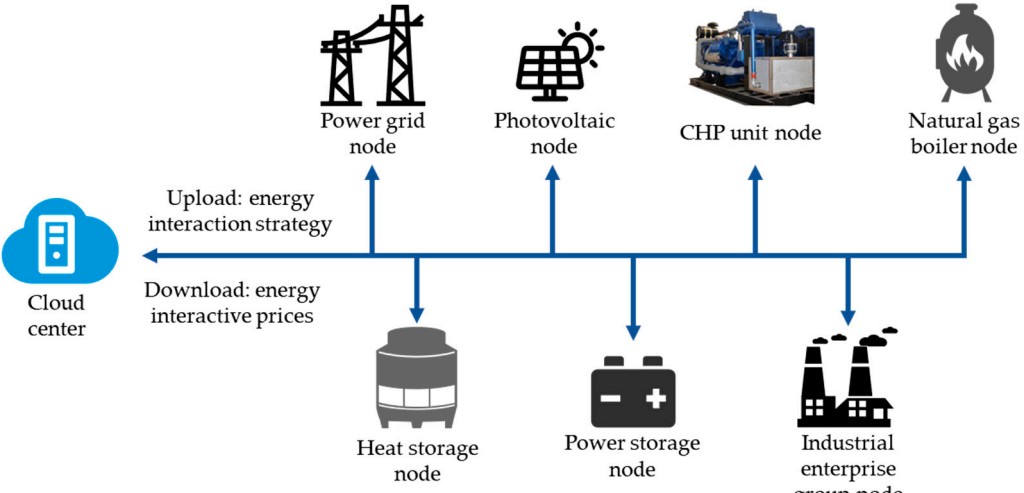

**Figure 2.** Industrial park edge–cloud information interaction mechanism.

Under the proposed interaction mechanism, each energy system node performs local optimization based on its operating status and the energy interactive price information issued by the cloud center. Subsequently, each node uploads the energy interaction strategy obtained by local optimization to the cloud center. The cloud center coordinates the energy interaction strategies of each node and updates the energy interaction price issuance. This model can greatly reduce the calculation amount at the cloud center and the demand for information from each node of the system. After obtaining the information from all nodes in the system, the cloud center calculates and updates the price information of all energy interactions in the system according to Formulas (48)–(50) and issues the information. Furthermore, an edge–cloud collaborative iterative optimization model is formed in which each edge node optimizes local independent optimization calculations and the cloud center collaboratively optimizes. In this mode, all nodes do not need to rely on a specific order for iterative solutions, and the cloud center can accept synchronous or asynchronous node information for collaborative optimization. The specific process of the algorithm is shown in Algorithm 1.

---

**Algorithm 1** Distributed iterative optimization algorithm based on edge–cloud collaboration

---

**Step 1:** Initialize the energy interaction price $\lambda$, $\mu$. Let the iteration number mark $r = 1$
**Step 2:** Each node completes local optimization calculations based on the current interaction price and uploads the calculated energy interaction strategy to the cloud center.
**Step 3:** Algorithm termination test. If the interaction price satisfies Formula (49) or the number of algorithm iterations reaches the upper limit, go to step 5; otherwise, go to step 4. In Formula (49), $\varepsilon$ is the accuracy threshold:

$$|\lambda_{r+1} - \lambda_r| + |\mu_{r+1} - \mu_r| \leq \varepsilon \tag{49}$$

**Step 4:** The cloud center updates the energy interaction price according to the energy interaction strategy of each node and Formulas (50)–(54). Among these, Formulas (50) and (51) calculate the update directions $g_{r,k}^{\mu,m\rightarrow n}$, $g_{r,k}^{\mu,m\rightarrow n}$ of the energy interaction price, Formula (52) determines the range of the update step $s_r$ of the energy interaction price, and Formulas (53) and (54) update the energy interaction price. $\phi^*$ is the optimal value of the overall model, which cannot be obtained accurately here. It can be estimated via the method in [35]. $L_m^*(\lambda, \mu)$ represents the optimal value of the m-node model under the energy interaction price $\lambda$, $\mu$.

$$g_{r,k}^{\mu,m\rightarrow n} = e_k^{n\leftarrow m} - w^{m\rightarrow n} e_k^{m\rightarrow n} \tag{50}$$

$$g_{r,k}^{\mu,m\rightarrow n} = q_k^{n\leftarrow m} - f^{m\rightarrow n} q_k^{m\rightarrow n} \tag{51}$$

$$0 < s_r < \frac{[\phi^* - \sum_{m\in M} L_m^*(\lambda, \mu)]}{\|\mathbf{g}_r\|^2} \tag{52}$$

$$\lambda_{k,r+1}^{m\rightarrow n} = \lambda_{k,r}^{m\rightarrow n} + s_r g_{r,k}^{\lambda,m\rightarrow n} \tag{53}$$

$$\mu_{k,r+1}^{m\rightarrow n} = \mu_{k,r}^{m\rightarrow n} + s_r g_{r,k}^{\mu,m\rightarrow n} \tag{54}$$

---

**Step 5:** Evaluate the resulting policy accuracy. The algorithm ends.

---

According to the definition of marginal price [36], when each node model applies the optimal interactive marginal price for local optimization, the optimal operating strategy of the overall system can be obtained. The energy interaction price between nodes is used as the introduced Lagrange multiplier to complete the relaxation of the flow model, so that each node model can be solved independently. In addition, the inter-node energy interaction price represents the marginal cost of energy transmission balance between nodes, which has the same definition as the marginal price of energy interaction. Moreover, Algorithm 1 represents the update process of multipliers according to the subgradient method [37], which ensures that the algorithm can converge to the optimal energy interaction marginal price of the system and obtain the optimal operating strategy [38,39].

## 4. Case Study and Numerical Results

### 4.1. The Description of the Case Study

The example calculation test was carried out based on an actual industrial park in Xinjiang. This test considers the time-of-use electricity price in Xinjiang [40]: peak period (8:00~11:00, 19:00~24:00) RMB 0.916096/kWh; flat period (11:00~14:00, 16:00~19:00, 0:00~2:00) RMB 0.557325/kWh; the 8 h low period (2:00~8:00, 14:00~16:00) RMB 0.198553/kWh. The on-grid electricity price [41] is RMB 0.262/kWh, and the natural gas price is RMB 2.25/cubic meter [42]. Since the time-of-use price is given in one-hour time units and most energy system scheduling work uses one hour as the time stage [13,14], so the test period of the calculation example was a seven-day week and is divided into 168 periods, that is $K = 168$.

The industrial park consists of three industrial enterprises, a CHP unit station, a natural gas boiler, a photovoltaic power station with a peak output of 10,000 kw, a power storage station, and a hot water storage tank. The specific parameters of these devices are as follows. The rated load $P^c$ of the CHP unit is 25,000 kW. The upper limit $\overline{x^c}$ of the load factor of the CHP unit is 1. The unit parameters $a, b, c, d$ are 31613.3, 1552.4, 5913.6, 2816.4,

respectively. The upper and lower limits $\underline{V^b}$, $\overline{V^b}$ of the battery state of charge of the power storage node are 0.1 and 0.9, respectively. The upper limits $\overline{p^{bc}}$, $\overline{p^{bd}}$ of the charging and discharging power of the power storage node are 2500 kWh. The maximum power storage capacity $E$ of the power storage node is 5000 kWh. The power attenuation coefficient $\mu^b$ is 0.98. The charge and discharge coefficients $\alpha^{bc}$, $\beta^{bd}$ are both 0.95. The upper limit $\overline{V^h}$ of heat storage capacity of the heat storage node is 50,000 kWh. The heat attenuation coefficient $\mu^h$ is 0.99. The calorific value $H^{ng}$ of natural gas is 9.78 kW/cubic meter. The thermal energy conversion coefficient $\eta^{gb}$ of the natural gas boiler is 0.85. The electrical flow line loss $w^{m \rightarrow n}$ and heat flow line loss $f^{m \rightarrow n}$ from node to node are both 0.99 in this industrial park. Photovoltaic output data are shown in Figure 3.

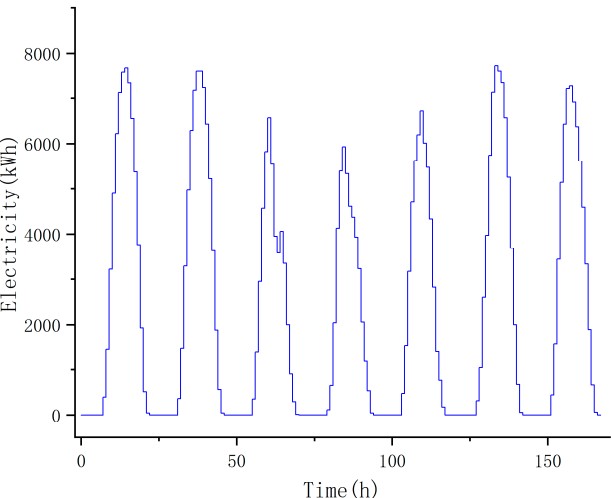

**Figure 3.** Photovoltaic output.

*4.2. System Optimization Operation Result*

Based on the above data, the supply and demand coordination optimization problem of the system is solved, and the obtained operation strategy is shown in Figures 4–7. Figure 4 shows the interaction results between the system and the power grid. The yellow column at the top represents the electric power purchased by the system from the power grid, and the green column at the bottom represents the electric power sold by the system to the power grid. Figure 5 shows the energy storage status of the power storage and heat storage nodes. The black line indicates the state of charge (SOC) level of the power storage node battery, and the red column indicates the amount of heat storage in the heat storage node. Figures 6 and 7, respectively, show the scheduling results for electric energy and thermal energy in industrial enterprises. The histograms of different colors represent the energy source nodes and sizes, respectively.

This paper takes the first day of operation as an example to illustrate the characteristics of operational strategy in the numerical test case. (1) In the low-load operational period for industrial enterprises (0–6 h), the load level on the demand side is low, and the power grid price is in a low period. It is not economical to turn on CHP at this time, so the power supply is mainly ensured by buying electricity during this period. (2) In the medium-load operational period for industrial enterprises (7 to 13 h), the level of electric heating load on the demand side is large, and the electricity price is basically flat. Therefore, there is good economy in operating the CHP, so the load is met by the grid to buy electricity and CHP. (3) At the peak of energy consumption by industrial enterprises (14–24 h), the time-of-use electricity price is at its highest, and most of the load is met by CHP and energy storage nodes. (4) During the operation of the system, both the storage node and the heat storage node store energy in the periods of medium or low electricity price and low load (0–13 h) and use this part of the energy when the electricity price is higher, to achieve demand response optimization.

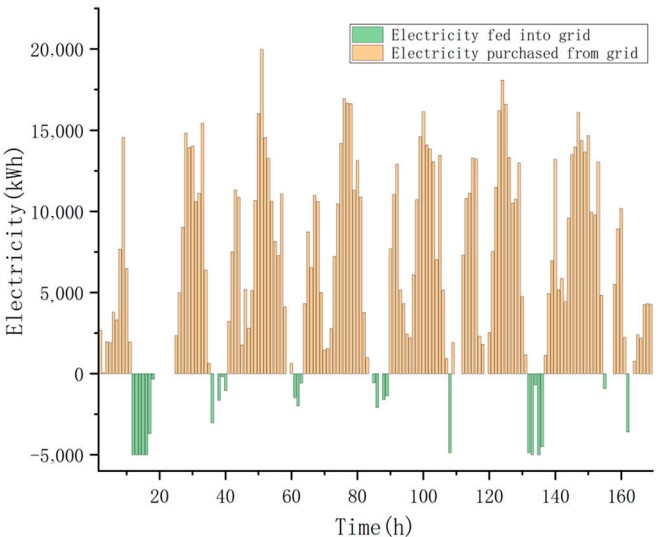

**Figure 4.** Result of interaction between system and power grid.

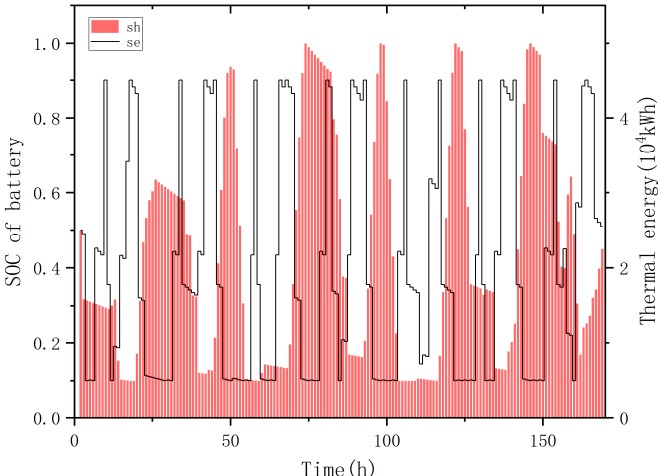

**Figure 5.** Scheduling results: energy storage.

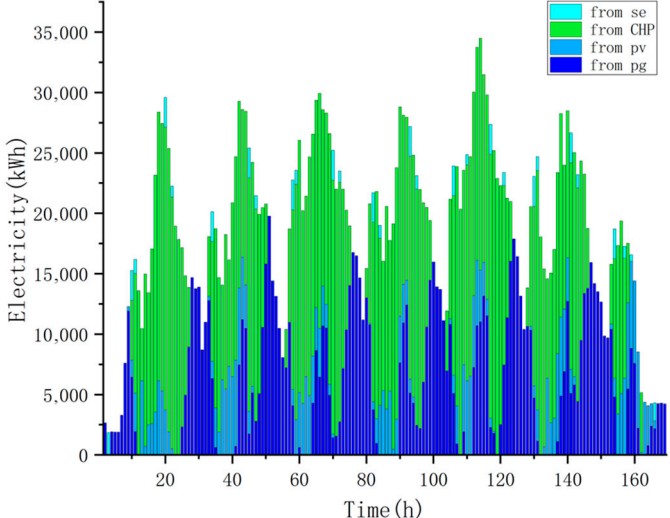

**Figure 6.** Electricity scheduling results: industrial enterprise.

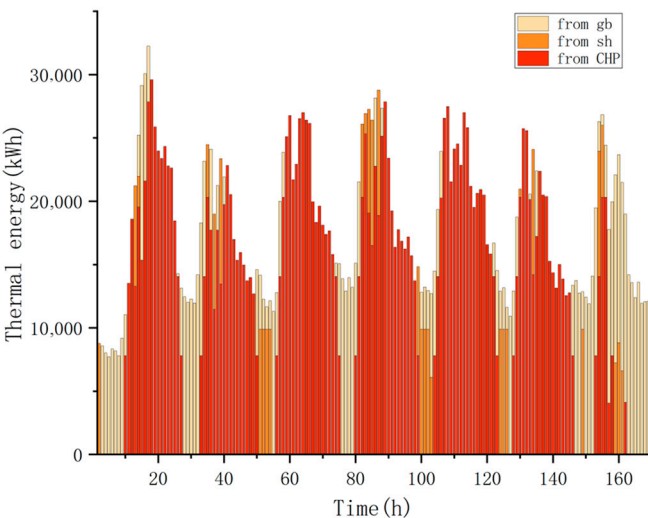

**Figure 7.** Thermal energy scheduling results: industrial enterprise.

In summary, through coordinated supply and demand dispatching for multiple industrial enterprises, and energy storage and supply equipment, the utilization efficiency of energy can be effectively improved, multi-energy synergy and complementation can be achieved, and the operating costs of integrated energy systems in industrial parks can be reduced.

### 4.3. Algorithm Performance Analysis

The algorithm tests were all conducted on a hardware platform with i7-10710U CPU and 16 G memory. In order to compare the performance of distributed algorithms, this study obtained the optimal solution to the centralized problem for performance analysis. The numerical test results obtained are shown in Table 2. The calculation formula of the gap between solutions is as follows:

$$Gap\ between\ solutions = \frac{Centralized\ solution\ result - Distributed\ solution\ result}{Centralized\ solution\ result} \tag{55}$$

**Table 2.** Comparison of case study results.

|  | System Operating Cost (RMB) | Solving Time (Seconds) |
|---|---|---|
| Centralized solution result | 2,312,480.03 | 9.29 |
| Distributed solution result | 2,354,024.75 | 4.46 |
| Gap between solution | −1.8% | 52.0% |

It can be seen that the proposed algorithm can shorten the solution time by more than 50% compared with solving centralized problems, and the gap between the result and the optimal solution is less than 2%, which meets the actual system operation requirements. Because the case tests used in this test had fewer nodes, the improvement effect is limited. However, the solution time of the distributed optimization algorithm proposed in this article will increase linearly as the number of problem nodes increases [43], so this method is suitable for actual large-scale multi-node integrated energy systems in industrial parks.

## 5. Conclusions and Future Work

China's industrial energy consumption is large. In response to the national strategic goal of "peak carbon neutrality" proposed by the Chinese government, it is urgent to improve energy efficiency in the industrial sector. This paper focuses on the supply and demand collaborative optimization scheduling problem of integrated energy systems in

industrial parks. The problem is jointly described by the node models and the flow model, and the "node-flow" model is also a mixed integer linear programming problem. Based on the "node-flow" model, an information interaction mechanism between edge nodes and the cloud is proposed to ensure the balance of energy transfer between nodes, and a distribution optimization algorithm is designed to optimize the operation strategy of each node in the industrial park to achieve efficient and optimized operation of the system. Numerical test results show that the proposed algorithm effectively improves the model's calculation efficiency within the range allowed by the accuracy of the scheduling strategy.

Further work can be conducted including the following three parts. First, in this paper, we consider only the operation scheduling problem of industrial park, but not the selection and capacity determination problem in the planning stages. The energy efficiency and economy of industrial parks can be improved by considering the operation problem and the planning problem together. Second, there is a lot of uncertainty regarding the process of the operation scheduling problem in industrial parks, so it is very meaningful to study this problem. The distributed iterative optimization algorithm proposed in this paper, combined with heuristic iterative algorithms [44] (such as the self-adaptive fast fireworks algorithm [45], adaptive polyploid memetic algorithm [46], and so on) to improve the speed and accuracy of problem solving, will be very valuable. For the large-scale multi-node optimization problem, the current algorithms [47,48] are still far from enough, and more in-depth research is needed.

**Author Contributions:** Conceptualization, G.L. and X.S.; methodology, G.L. and C.X.; writing—original draft preparation, C.X. and T.L.; writing—review and editing, T.L. and Y.L.; funding acquisition, G.L. and K.L. All authors have read and agreed to the published version of the manuscript.

**Funding:** This research is funded by Major Science and Technology Projects of Xinjiang Uygur Autonomous Region (2022A01001) and the National Natural Science Foundation of China (61903293).

**Institutional Review Board Statement:** Not applicable.

**Informed Consent Statement:** Not applicable.

**Data Availability Statement:** The data presented in this study will be made available on request.

**Conflicts of Interest:** Author Gengshun Liu is employed by Guoneng Xinjiang Ganquanbao Comprehensive Energy Co., Ltd. Authors Xinfu Song and Chaoshan Xin are employed by Economic and Technological Research Institute, State Grid Xinjiang Electric Power Co., Ltd. Author Gengshun Liu received a research grant from Major Science and Technology Projects of Xinjiang Uygur Autonomous Region (2022A01001). Author Kun Liu received a research grant from the National Natural Science Foundation of China (61903293). The remaining authors declare that the research was conducted in the absence of any commercial or financial relationships that could be construed as potential conflicts of interest.

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
