# Peer review of "Edge–Cloud Collaborative Optimization Scheduling of an Industrial Park Integrated Energy System"

_sustainability, doi:10.3390/su16051908_

Round 1

Reviewer 1 Report

Comments and Suggestions for Authors

The work is well done in terms of ideas and planning.

The bibliography is clear, the goals are well explained.

The conclusions are also comprehensive.

The problem is in the model which is not explained, not introduced, not validated, not tested and not accompanied by a bibliography.

It is difficult to understand the formulae and their application.

Furthermore, all the numerous acronyms and abbreviations in the formulae should be explained graphically in practical tables.

The formulae should be alternated with the explanation text.

The work certainly needs to be revised and better explained.

Reviewer 2 Report

Comments and Suggestions for Authors

The study presents a "node-flow" model that describes the coupling of multiple energy and information sources in an industrial park's integrated energy system. It proposes an edge-cloud information interaction mechanism for energy transfer balance between nodes and a distributed iterative optimization algorithm for optimizing each node's operation. The algorithm aims to enhance computational efficiency and precision in energy scheduling strategies. The research demonstrates that this approach effectively improves energy utilization efficiency, achieves multi-energy synergy, and reduces operating costs in industrial parks' integrated energy systems.

Point 1: The paper presents various equations integral to the model. It would be beneficial to include more detailed explanations or step-by-step derivations of these equations to enhance understanding.

Point 2: Elaborating on how these equations specifically contribute to the energy optimization goals would be valuable.

Point 3. Row 187, 213: check equation numbers.

Reviewer 3 Report

Comments and Suggestions for Authors

This paper concentrates on enhancing the efficiency of an integrated energy system with coordinated supply-demand management in an industrial park. The issue is conceptualized as a "node-flow" model, effectively capturing the interconnection of various energy and information aspects in the system, along with the interplay between the information and energy systems. Utilizing the node-flow model as a foundation, we introduce an edge-cloud information interaction mechanism that relies on maintaining an energy transfer balance between nodes. Additionally, a distributed iterative optimization algorithm is devised based on edge-cloud collaboration, aiming to achieve optimized decisions for each node. The outcomes demonstrate that the proposed algorithm significantly enhances the computational efficiency of the model within the precision limits of the scheduling strategy. However, it has some minor flaw commented below:

How did authors decide to divide calculation example of seven days into 168 periods? Is it based on different electricity price in different periods? Can further explanation be given?

Reviewer 4 Report

Comments and Suggestions for Authors

In this work, the main researches focus on the optimization of integrated energy system with supply-demand coordination in industrial park. The research topic is interesting and important. However, there are many problems in the paper, and it is recommended to resubmit it after revision, the specific comments are as follows:

1)     The abstract is repetition and confusion, and it is recommended to revise it carefully and add quantitative conclusions.

2)     The structure of the manuscript is recommended to be rewritten with reference to the journal template. Meanwhile, the language of the manuscript suggests further polishing and modification.

3)     Sections 2.1-2.7 give different types of node models optimization method and list calculation formulas, but how to implement them is still unclear.

4)     Suggest to give a detailed description of the analysis engineering case in section 4.1, including the system unit composition, parameters and dynamic changes of energy flow.

5)     What is the special function of PV output data in Figure 3? Suggest to delete.

6)     Figures 4-7 lack a detailed and in-depth analysis, and it is recommended to add quantitative comparison data.

7)     In Line 373, “the error between the result and the optimal solution is less than 2%,” How did this conclusion come about?

Comments on the Quality of English Language

The language of the manuscript suggests further polishing and modification.

Round 2

Reviewer 4 Report

Comments and Suggestions for Authors

All questions are answered correctly, suggest to accept the paper.

Comments on the Quality of English Language

none

Author Response

Thank for comments!